# Urinary metabolic biomarkers of diet quality in European children are associated with metabolic health

Nikos Stratakis[1]*[†], Alexandros P Siskos[2†], Eleni Papadopoulou[3†], Anh N Nguyen[4], Yinqi Zhao[1], Katerina Margetaki[1], Chung-Ho E Lau[2,5], Muireann Coen[2,6], Lea Maitre[7,8,9], Silvia Fernández-Barrés[7,8,9], Lydiane Agier[10], Sandra Andrusaityte[11], Xavier Basagaña[7,8,9], Anne Lise Brantsaeter[3], Maribel Casas[7,8,9], Serena Fossati[7,8,9], Regina Grazuleviciene[11], Barbara Heude[12], Rosemary RC McEachan[13], Helle Margrete Meltzer[3], Christopher Millett[14,15], Fernanda Rauber[14,15,16], Oliver Robinson[5], Theano Roumeliotaki[17], Eva Borras[7,18], Eduard Sabidó[7,18], Jose Urquiza[7,8,9], Marina Vafeiadi[17], Paolo Vineis[5], Trudy Voortman[4], John Wright[13], David V Conti[1], Martine Vrijheid[7,8,9], Hector C Keun[2‡], Leda Chatzi[1‡]

[1]Department of Preventive Medicine, Keck School of Medicine, University of Southern California, Los Angeles, United States; [2]Cancer Metabolism & Systems Toxicology Group, Division of Cancer, Department of Surgery & Cancer and Division of Systems Medicine, Department of Metabolism, Digestion & Reproduction, Imperial College London, Hammersmith Hospital Campus, London, United Kingdom; [3]Norwegian Institute of Public Health, Oslo, Norway; [4]Department of Epidemiology, Erasmus University Medical Center, Rotterdam, Netherlands; [5]MRC Centre for Environment and Health, School of Public Health, Imperial College London, London, United Kingdom; [6]Oncology Safety, Clinical Pharmacology and Safety Sciences, R&D, AstraZeneca, Cambridge, United Kingdom; [7]ISGlobal, Barcelona, Spain; [8]Universitat Pompeu Fabra, Barcelona, Spain; [9]CIBER Epidemiologia y Salud Pública, Madrid, Spain; [10]Inserm, CNRS, University Grenoble Alpes, Team of environmental epidemiology applied to reproduction and respiratory health, IAB, Grenoble, France; [11]Department of Environmental Sciences, Vytautas Magnus University, Kaunas, Lithuania; [12]Centre for Research in Epidemiology and Statistics, Université de Paris, Inserm, Inra, Paris, France; [13]Bradford Institute for Health Research, Bradford Teaching Hospitals NHS Foundation Trust, Bradford, United Kingdom; [14]Public Health Policy Evaluation Unit, School of Public Health, Imperial College, London, United Kingdom; [15]Department of Preventive Medicine, School of Medicine, University of São Paulo, São Paulo, Brazil; [16]Center for Epidemiological Research in Nutrition and Health, University of São Paulo, São Paulo, Brazil; [17]Department of Social Medicine, Faculty of Medicine, University of Crete, Heraklion, Greece; [18]Centre for Genomic Regulation, The Barcelona Institute of Science and Technology, Barcelona, Spain

*For correspondence:
nstratak@usc.edu

[†]These authors contributed equally to this work
[‡]These authors also contributed equally to this work

**Abstract** Urinary metabolic profiling is a promising powerful tool to reflect dietary intake and can help understand metabolic alterations in response to diet quality. Here, we used [1]H NMR spectroscopy in a multicountry study in European children (1147 children from 6 different cohorts) and identified a common panel of 4 urinary metabolites (hippurate, N-methylnicotinic acid, urea, and sucrose) that was predictive of Mediterranean diet adherence (KIDMED) and ultra-processed food consumption and also had higher capacity in discriminating children's diet quality than that

of established sociodemographic determinants. Further, we showed that the identified metabolite panel also reflected the associations of these diet quality indicators with C-peptide, a stable and accurate marker of insulin resistance and future risk of metabolic disease. This methodology enables objective assessment of dietary patterns in European child populations, complementary to traditional questionary methods, and can be used in future studies to evaluate diet quality. Moreover, this knowledge can provide mechanistic evidence of common biological pathways that characterize healthy and unhealthy dietary patterns, and diet-related molecular alterations that could associate to metabolic disease.

## Editor's evaluation

This well executed study looks at the association of urinary metabolites to the types of diets consumed by European children. Using NMR they find four metabolites that are predictive of a Mediterranean diet. This presents both an approach additional to traditional questionnaire methods and potential insights into biological pathways and will be of interest to nutritionists and epidemiologists.

## Introduction

Dietary habits are considered a key element for the prevention of chronic noncommunicable diseases (*Grosso et al., 2020*). In 2017, 22% of all deaths among adults were attributed to dietary risks with type 2 diabetes being among the top causes of diet-related deaths (*Collaborators, 2019*). However, it is notoriously difficult to measure diet accurately in large population studies and the absence of accurate dietary assessment methods is hampering the evidence linking diet and disease (*Posma et al., 2020*; *Ioannidis, 2018*). There is a need for novel approaches to better elucidate diet-related metabolic alterations and their association with disease risk.

Metabolomics is the systematic study of small-molecule metabolites in a biological system and has recently emerged as a powerful top-down approach providing a comprehensive phenotype of biological status. Urinary metabolic phenotypes carry rich information on environmental, lifestyle and nutritional exposures, physiological and metabolic status, and disease risks on an individual and population level (*Gibson et al., 2020*; *Collins et al., 2019*; *Rebholz et al., 2018*). Urine specimens have high concentrations of food-derived metabolites and studies have shown that urinary metabolic profiles could provide an objective measure of dietary intake (*O'Gorman and Brennan, 2017*).

Previous research identifying diet-related metabolic profiles has largely focused on selected food groups including fruits, vegetables, meat, and seafood (*Gibson et al., 2020*; *Lau et al., 2018*; *Guertin et al., 2014*; *Playdon et al., 2016*; *Scalbert et al., 2014*), while dietary patterns and food processing are far less studied (*Collins et al., 2019*; *Rebholz et al., 2018*; *Garcia-Perez et al., 2017*; *Martinez et al., 2016*). Ultra-processed foods (UPFs), which are industrial formulations undergoing a series of physical and chemical processes and typically lack intact healthy food components and include various additives, can result to cumulative intake of salt, added sugars, and fats (*Monteiro et al., 2019*). UPF consumption has been increasing worldwide (*Monteiro et al., 2013*; *Monteiro et al., 2018a*; *Monteiro et al., 2018b*; *Baker et al., 2020*) and, to our knowledge, no previous report exists on its metabolic signature. In contrast with the study of overall diets, including the UPF diet, exploring intakes of single foods, which is what is traditionally done in nutrition research, might be failing to provide a realistic image of dietary metabolic footprint. Moreover, most previous studies have focused on adults, and little is known about body's metabolic response to diet during childhood. This is important as age is a major source of variation in metabolite profiling, with large differences being observed between children and adults (*Ellul et al., 2019*).

Alterations in metabolite profiling could provide a mechanistic link between diet and disease development as early as in childhood. Alterations in the levels of several metabolic biomarkers and pathways have been associated with insulin resistance in childhood including branched-chain and aromatic amino acid metabolism, urea cycle, glucose, and carbohydrate metabolism (*Rauschert et al., 2017*; *Zhao et al., 2016*; *Martos-Moreno et al., 2017*). In addition, dietary patterns characterized by high consumption of UPFs, such soft drinks, sweet, and savoury snacks, have been related to higher risk of insulin resistance in children (*Karatzi et al., 2014*; *Romero-Polvo et al., 2012*). All previous studies

have included measurement of serum insulin levels as marker of insulin resistance. An alternative marker is C-peptide, a protein that is cosecreted with insulin on an equimolar basis from pancreatic β-cells and has been shown to strongly predict metabolic disease progression (*Patel et al., 2012*). C-peptide has a longer half-life than insulin and is recognized as a stable and accurate marker of endogenous insulin secretion, even in nonfasting conditions (*Hope et al., 2016*; *Vezzosi et al., 2007*; *Polonsky et al., 1986*). Previous studies in children have shown that higher carbohydrate intake is related to higher C-peptide concentrations (*Sunehag et al., 2002*; *Buyken et al., 2006*). However, there is little evidence on the metabolic signatures underlying the association of diet with C-peptide levels in children.

We conducted a multicountry study in European children within the Human Early-Life Exposome (HELIX) project (*Maitre et al., 2018*) aiming (1) to identify urinary metabolites associated with Mediterranean diet adherence and UPF consumption, and (2) to determine the extent to which these metabolites were associated with C-peptide, used as an early marker of metabolic health.

## Methods
### Study population
This study is embedded within the HELIX project (*Maitre et al., 2018*), a collaborative project across six established and ongoing longitudinal population-based birth cohort studies in Europe: Born in Bradford (BiB, UK) (*Wright et al., 2013*), Étude des Déterminants pré et postnatals du développement et de la santé de l'Enfant (EDEN, France) (*Heude et al., 2016*), Kaunas Cohort (KANC, Lithuania) (*Grazuleviciene et al., 2009*), INfancia y Medio Ambiente (INMA, Spain) (*Guxens et al., 2012*), Norwegian Mother, Father and Child Cohort Study (MoBa, Norway) (*Magnus et al., 2016*), and RHEA (RHEA, Greece) (*Chatzi et al., 2017a*). Participating cohorts covered singleton deliveries from 2003 to 2008. As part of HELIX, a subcohort of 1301 children (approximately 200 children in each cohort) were followed in 2014–2015 for a clinical examination, a computer-assisted interview with the parents, and the collection of biological samples. Data collection was standardized across cohorts and performed by trained staff. A full description of the HELIX follow-up methods and study population are provided by *Maitre et al., 2018*. Prior to the start of HELIX, all six cohorts on which HELIX is based had

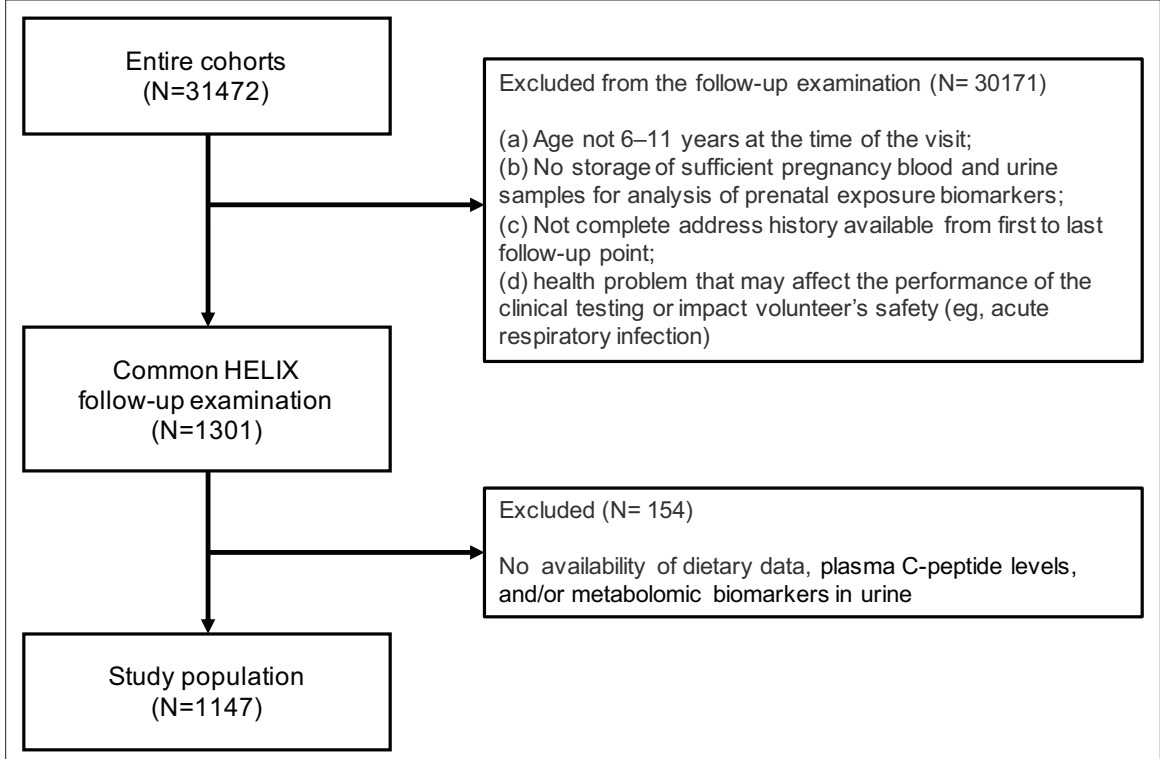

**Figure 1.** Participant flowchart.

undergone the required evaluation by national ethics committees and had obtained all the required permissions for their cohort recruitment and follow-up visits. Each cohort also confirmed that relevant informed consent and approval were in place for secondary use of data from preexisting data. The work in HELIX was covered by new ethics approvals from the local ethics committees at each site, and at enrolment in the HELIX subcohort, participants were asked to sign an informed consent form for the specific HELIX work including clinical examination and biospecimen collection and analysis. Additionally, the current study was approved by the University of Southern California Institutional Review Board.

Our study population consisted of 1147 children with available information on dietary intake, plasma C-peptide levels, and metabolomic biomarkers in urine collected during the HELIX follow-up at a mean age of 7.9 years (range: 5.4–12.0 years) (*Figure 1*).

## Dietary assessment

Information about the children's habitual diet was collected via a semi quantitative food-frequency questionnaire (FFQ) covering the child's habitual diet, which was filled in by the parent attending the examination appointment. The FFQ, covering the past year, was developed by the HELIX research group, translated and applied to all cohorts (*Maitre et al., 2018*). It included 43 questions of intake of food items, which were aggregated in 16 main food groups: meat and meat products, fish and seafood, sweets, beverages, potatoes, vegetables, dairy products, fruits, bread and cereal, sweet bakery products, added fats, eggs, nuts, salty snacks, pulses, and dressings. It also included 15 specific questions to examining the degree of adherence to Mediterranean diet. Diet quality was assessed using two different approaches, (1) by assessing the degree of adherence to a Mediterranean diet based on the KIDMED index (Mediterranean Diet Quality Index for children and adolescents) (*Serra-Majem et al., 2004*) and (2) by assessing the proportion of UPF in the overall diet (*Monteiro et al., 2019*).

For the KIDMED index (*Serra-Majem et al., 2004*), items positively associated with the Mediterranean diet pattern (11 items) were assigned a value of +1, while those negatively associated with the Mediterranean diet pattern (4 items) were assigned a value of −1 (*Supplementary file 1a*). The scores for all 15 items were summed, resulting in a total KIDMED score ranging from −4 to 11, with higher scores reflecting greater adherence to a Mediterranean diet. We categorized the score into three groups: low (<1), moderate (1–4), and high (>4).

For UPF intake, we identified foods and drinks as 'ultra-processed' by using the NOVA classification, a food classification system based on the nature, extent, and purpose of industrial food processing (*Monteiro et al., 2019*) We identified the following 'ultra-processed' foods: cookies, pastries, sugar-sweetened, low-sugar and artificially sweetened beverages, cold meat cuts; ham, dairy desserts, sugar-sweetened and other breakfast cereals, crispbread and rusks; chocolate, sweets, margarine, dressings, and salty snacks. For some food items, our FFQ did not provide enough information on food processing to determine if a specific item belongs to one processing category or another. We discussed the classification of each food item with a team of nutritionists and used a conservative approach, such that the lower level of processing was chosen – for instance, we made the assumption that fries are homemade from fresh potatoes, and therefore, they were not classified as UPF. For each child, we calculated the daily proportion of all UPF in the total diet as the ratio between the sum of daily servings of UPF to the total daily sum of all food and drink servings. More details on the categorization of foods according to the NOVA classification are presented in *Supplementary file 1b*.

## Urine metabolite profiling

Two urine samples, representing last night-time and first morning voids, were collected on the evening and morning before the clinical examination, kept in a fridge and transported in a temperature-controlled environment, and aliquoted and frozen within 3 hr of arrival at the clinics. They were subsequently pooled to generate a more representative sample of the last 24 hr for metabolomic analysis (*Lau et al., 2018*).

Urinary metabolic profiles were acquired using $^1$H NMR spectroscopy according to *Lau et al., 2018*. In brief one-dimensional 600 MHz $^1$H NMR spectra of urine samples from each cohort were acquired on the same Bruker Avance III spectrometer operating at 14.1 Tesla within a period of 1 month. The spectrometer was equipped with a Bruker SampleJet system, and a 5-mm broadband inverse configuration probe maintained at 300 K. Prior to analysis, cohort samples were randomized. Deuterated

3-(trimethylsilyl)-[2,2,3,3-d$_4$]-propionic acid sodium salt was used as internal reference. Aliquots of the study pooled quality control (QC) sample were used to monitor analytical performance throughout the run and were analysed at an interval of every 23 samples (i.e., 4 QC samples per well plate). The $^1$H NMR spectra were acquired using a standard one-dimensional solvent suppression pulse sequence. Forty-four metabolites were identified and quantified as described in *Lau et al., 2018*. The urinary NMR showed excellent analytical performance, the mean coefficient of variation across the 44 NMR detected urinary metabolites was 11%. For the statistical analysis we have used creatinine-normalized metabolite concentrations (µmol/mmol of creatinine).

## Plasma C-peptide

Blood was collected at the end of the clinical examination during the HELIX follow-up visit. The median postprandial interval (time between last meal and blood collection) was 3.3 hr (interquartile range [IQR: 2.8–4.0]).

For each cohort, concentration of C-peptide was assessed in child plasma at the CRG/UPF Proteomics Unit (Barcelona, Spain) using the xMAP and Luminex System multiplex platform according to the manufacturer's protocol. Blood samples were randomized and blocked by cohort prior to measurement to ensure a representation of each cohort in each plate (batch). For protein quantification, an 8-point calibration curve per plate was performed with protein standards provided in the Luminex kit and following the procedures described in the standard procedures described by the vendor. Commercial heat inactivated, sterile-filtered plasma from human male AB plasma (Sigma Cat # H3667) was used as constant controls to control for intra- and interplate variability. Four control samples were added per plate. Raw intensities obtained with the xMAP and Luminex system for each sample were converted to pg/ml using the calculated standard curves of each plate and accounting for the dilutions that were made prior measurement. The coefficient of variation for C-peptide was 16%. The LOD was determined and the lower and upper quantification limits (LOQ1 and LOQ2, respectively) were obtained from the calibration curves. C-peptide concentrations were log2-transformed to achieve normal distribution. Plate batch effect was corrected by subtracting for each individual and each protein the difference between the overall protein average minus the plate-specific protein average. Finally, values below LOQ1 and above LOQ2 were imputed using a truncated normal distribution using the truncdist R package.

## Covariates

Adjustment factors were selected a priori based on literature (*Aranceta et al., 2003*; *Scaglioni et al., 2018*; *Patrick and Nicklas, 2005*) and included: maternal age (in years), maternal education level (low, middle, high), maternal prepregnancy body mass index (BMI, in kg/m$^2$), family affluence score (cohort-specific definition of low, middle, high), child sex, child age (in years), child BMI (in kg/m$^2$), child sedentary behavior (min/day of time spent watching TV, playing computer games or other sedentary games), child ethnicity (White European, Asian, other), and postprandial interval (in hours). We also included a cohort indicator as a fixed effect in the models, as this, in the context of an observational study, is expected to control for cohort effects (*Basagaña et al., 2018*). We imputed missing values for covariates (ranging from 0% to 4%) using the method of chained equations with the R package *mice*. Details about the imputation process in HELIX, diagnostics, and comparison between imputed and complete-case values have been reported in detail elsewhere (*Agier et al., 2019*).

## Statistical analysis

As a first step in our analysis, we conducted a metabolome-wide association study to assess the associations of urinary metabolites with diet quality. Creatinine-normalized metabolite concentrations (µmol/mmol of creatinine) were log$_{10}$ transformed prior to statistical analyses to improve model fit. We fitted separate multivariable regression models for each metabolite with the KIDMED score or UPF intake (expressed as per 5% change of total daily food intake). To account for multiple hypothesis testing, we applied the Benjamini–Hochberg false discovery rate (FDR) correction; an FDR-corrected p value <0.05 denoted statistical significance. For metabolites identified to be associated with the diet quality indicators, we assessed between-cohort heterogeneity with the $I^2$ statistic and $\chi^2$ test from Cochran's Q.

To examine the ability of the identified metabolite panels in discriminating children with low vs. high KIDMED scores (<1 vs. >4) and low vs. high UPF intakes (Quartile 1: <18% vs. Quartile 4: ≥29%), we plotted receiver operating characteristic (ROC) curves and estimated area under the ROC curve values, indicative of the discriminative performance of the metabolite models, based on tenfold cross validations. We also repeated the ROC analysis for a set of established sociodemographic factors (maternal education level, maternal prepregnancy BMI, family affluence score, child sedentary behavior, ethnicity, age, and sex) linked to childhood diet quality both previously (*Aranceta et al., 2003*; *Scaglioni et al., 2018*; *Patrick and Nicklas, 2005*) and in our study population (all p < 0.05) and compared the discriminative performance of this sociodemographic set with that of the metabolites.

Next, we examined the association of the KIDMED score and of UPF intake (as independent variables) with plasma C-peptide concentration (as dependent variable) using multivariable linear regression models. No departures from linearity in the associations of these diet quality indicators with C-peptide concentration were observed both visually and statistically (p for linearity >0.42) using generalized additive models. We examined the KIDMED score both as continuous (per score unit increase) and in categories of low (score <1, reference), moderate (score = 1–4), and high (score >4). Likewise, UPF intake was assessed both as continuous (per 5% change of total daily food intake) and in quartiles (Q1: <18%, reference; Q2: 18% to <23%; Q3: 23% to <29%; and Q4: ≥29% of total daily food intake). We also included a product term between KIDMED and UPF intake in the regression analysis to assess their interaction; to simplify interpretation of this model, we categorized the KIDMED score as low/moderate vs. high and UPF intake based on the median population intake ( <23% vs. ≥23%). We conducted two sets of sensitivity analyses to assess the robustness of the results. First, we calculated cohort-specific effect estimates and assessed heterogeneity with the $I^2$ statistic and $\chi^2$ test from Cochran's Q. Second, we examined potential effect modification by child sex and by child weight status (IOTF-defined normal weight vs. overweight/obese) on C-peptide by testing the multiplicative interaction term between the potential effect modifier and each diet quality measure. Finally, we fitted regression models with the metabolites found to be associated

**Table 1.** Characteristics of the study population.

| Cohort of inclusion, *n* (%) | |
|---|---|
| BiB, UK | 189 (16.5) |
| EDEN, France | 149 (13) |
| INMA, Spain | 202 (17.6) |
| KANC, Lithuania | 194 (16.9) |
| MoBa, Norway | 221 (19.3) |
| RHEA, Greece | 192 (16.7) |
| **Maternal characteristics** | |
| Maternal age, mean (SD), years | 30.7 (4.9) |
| Missing, *n* (%) | 13 (1.1) |
| Prepregnancy BMI, mean (SD), kg/m$^2$ | 25 (5) |
| Missing, *n* (%) | 21 (1.8) |
| Maternal educational level, *n* (%) | |
| Low | 157 (13.7) |
| Medium | 391 (34.1) |
| High | 562 (49) |
| Missing, *n* (%) | 37 (3.2) |
| **Child characteristics** | |
| Age at assessment, mean (SD), years | 7.9 (1.6) |
| Sex, *n* (%) | |
| Male | 626 (54.6) |
| Female | 521 (45.4) |
| Ethnicity, *n* (%) | |
| White European | 1,028 (89.6) |
| Asian | 92 (8) |
| Other | 27 (2.4) |
| Family affluence score, *n* (%) | |
| Low | 126 (11) |
| Medium | 448 (39.1) |
| High | 569 (49.6) |
| Missing, *n* (%) | 4 (0.4) |
| BMI, mean (SD), kg/m$^2$ | 16.9 (2.6) |
| Normal weight, *n* (%)* | 906 (79) |
| Overweight/obese, *n* (%)* | 237 (20.7) |
| Missing, *n* (%) | 4 (0.4) |
| KIDMED score, mean (SD) | 2.8 (1.7) |
| Low (<1), *n* (%) | 104 (9.1) |
| Medium (1–4), *n* (%) | 848 (73.9) |
| High (>4), *n* (%) | 195 (17) |

*Table 1 continued on next page*

*Table 1 continued*

Cohort of inclusion, **n (%)**

| | |
|---|---|
| Ultra-processed food intake, mean (SD), % of daily food intake | 24.2 (8.7) |

*Categories of normal weight and overweight/obese were derived using the International Obesity Taskforce criteria (**Cole and Lobstein, 2012**).

BiB = Born in Bradford cohort. EDEN = the Étude des Déterminants pré et postnatals du développement et de la santé de l'Enfant study. INMA = INfancia y Medio Ambiente cohort. KANC = Kaunas Cohort. KIDMED = Mediterranean Diet Quality Index for children and adolescents. MoBa = Norwegian Mother, Father and Child Cohort Study. RHEA = Rhea Mother Child Cohort study.

with diet quality indicators, and C-peptide in order to assess whether diet-related metabolites are related to β-cell function.

We performed analyses with both complete (missingness <4% in each covariate) and imputed data. Results were similar across raw and imputed data analyses, and hence, we present those using the imputed covariate data. For easier interpretation of effect estimates for the log-transformed C-peptide and metabolite values, we back-transformed regression coefficients and present results as percent change (% change = (back-transformed [beta] − 1) × 100).

Analyses were conducted using STATA version 14.2 (StataCorp LLC, TX) and R software version 3.5.3. Linear regression analyses were performed in STATA with the command '*regress*' and using '*mi estimate*' to account for the imputed covariate data (**StataCorp, 2021**). ROC analysis was performed in R with the caret package (**Kuhn, 2008**). Visualizations of the results were carried out using the ggplot2 package in R (**Wickham, 2016**).

## Results
### Study population
Among participating children (*n* = 1147), 626 (54.6%) were boys and 1028 (89.6%) were white (**Table 1**). The mean (SD) age at assessment was 7.9 (1.6) years. Median (IQR) C-peptide concentration was 1.26 (0.03, 1.95) ng/ml.

Seventeen percent of children (*n* = 195) had a high KIDMED score (>4), indicative of high adherence to the Mediterranean diet. Mean (SD) UPF intake in the overall study population was 24.2 (8.7)% of total daily food intake. KIDMED and UPF intake were negatively correlated (Spearman *r* = −0.44); children with high and low (<1) KIDMED score had a mean (SD) UPF intake of 18.8 (6.6)% and 33.4 (9.9)%, respectively. The two diet quality scores were associated with most recorded food intakes in opposite directions (**Supplementary file 1c, d**). Daily fruit and vegetable consumption, weekly fish consumption and not skipping breakfast were the major dietary habits differentiating children with low and high KIDMED score (**Supplementary file 1c**). Intakes of pastries and bakery products, dairy

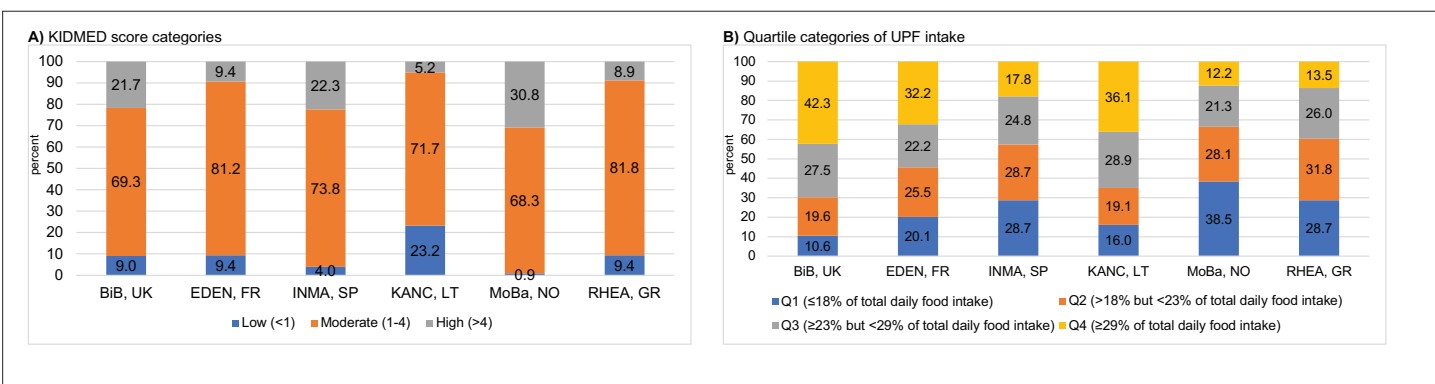

**Figure 2.** Levels of childhood adherence to the diet quality indicators of interest in each Human Early-Life Exposome (HELIX) subcohort. Panel (**A**) illustrates the levels of adherence to the Mediterranean diet which were defined as follows: low, KIDMED score, <1; moderate, KIDMED score, 1–4; and high, >4. Panel (**B**) illustrates the levels of ultra-processed food consumption (expressed as % of total daily food intake) which are based on quartile (*Q*) cutoffs according to the intake distribution of the overall HELIX study population. BiB, Born in Bradford cohort; EDEN, the Étude des Déterminants pré et postnatals du développement et de la santé de l'Enfant study; INMA, INfancia y Medio Ambiente cohort; KANC, Kaunas Cohort; KIDMED, Mediterranean Diet Quality Index for children and adolescents; MoBa, Norwegian Mother, Father and Child Cohort Study; RHEA, Rhea Mother Child Cohort study.

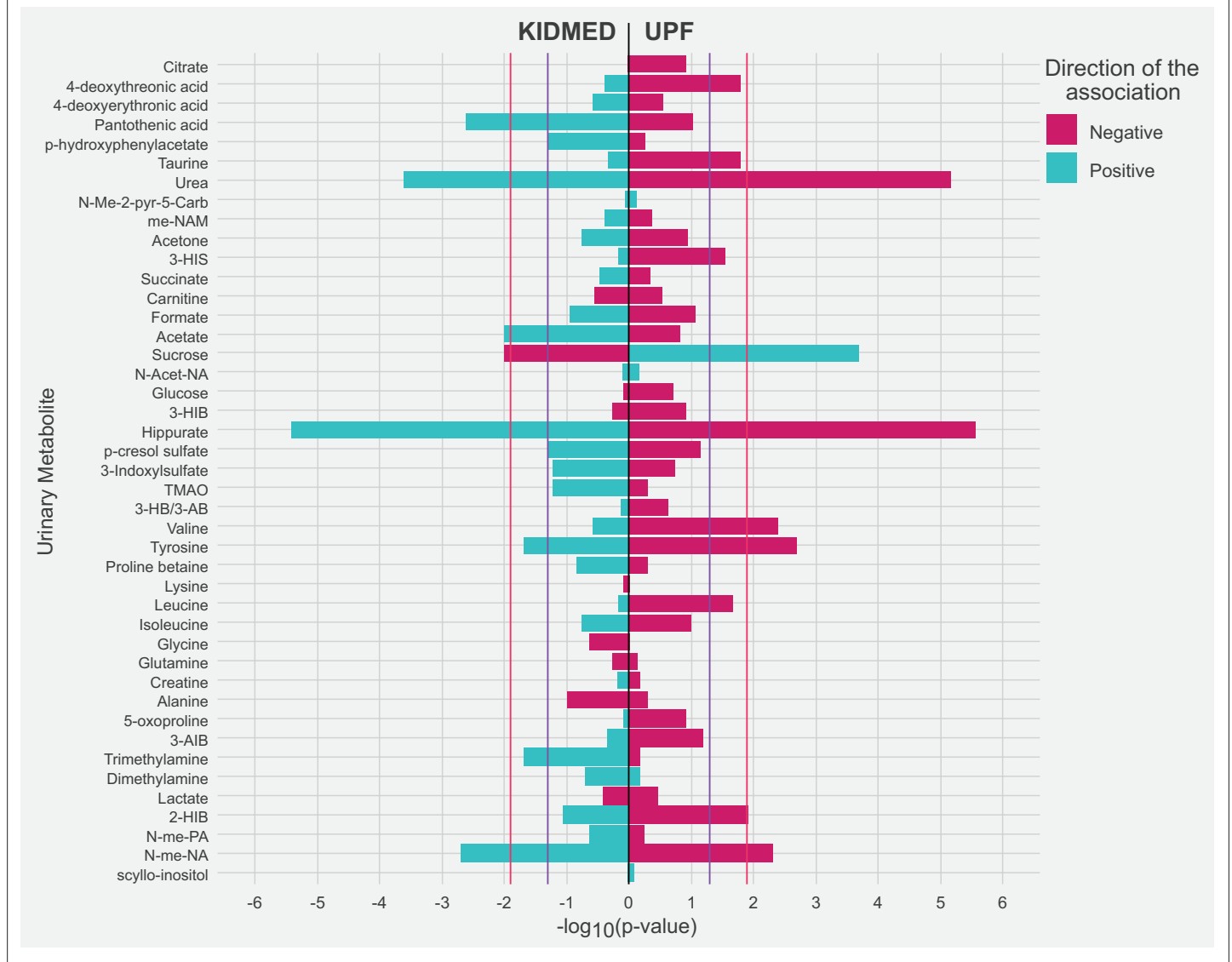

**Figure 3.** Adjusted associations of the diet quality indicators of interest with urinary metabolites in childhood. Linear regression models were adjusted for maternal age, maternal education level, maternal prepregnancy body mass index (BMI), family affluence status, child sex, child age, child BMI, child sedentary behavior, child ethnicity, and a cohort indicator. The purple line represents a p value of 0.05. The red line represents an false discovery rate (FDR)-adjusted p value of 0.05. 2-HIB, 2-hydroxyisobutyrate; 3-AIB, 3-aminoisobutyrate; 3-HB/3-AB, 3-hydroxybutyrate/3-aminoisobutyrate; 3-HIB, 3-hydroxyisobutyrate; 3-HIS, 3-hydroxyisovalerate; me-NAM, N1-methyl-nicotinamide; N-Acet-NA, N-acetyl neuraminic acid; N-Me-2-pyr-5-Carb, N-methyl-2-pyridone-5-carboxamide; N-me-NA, N-methylnicotinic acid; N-me-PA, N-methylpicolinic acid; TMAO, trimethylamine N-oxide.

desserts, margarine, and dressings were the major determinants of UPF intake (*Supplementary file 1d*). Children with the highest KIDMED scores were mostly from Norway and Spain (*Figure 2A*), while those with the highest UPF intake were mostly from Lithuania and the UK (*Figure 2B*).

## Diet quality and urinary metabolome

*Figure 3* and *Supplementary file 1e, f* show the associations of diet quality indicators with urinary metabolite levels in childhood. KIDMED and UPF intake exhibited an opposite pattern of association for most of the metabolites (30 out of 43). After controlling for FDR (FDR-corrected p value <0.05), we found that a panel of four metabolites related to both diet quality indicators. Specifically, a higher KIDMED score was associated with higher levels of hippurate, *N*-methylnicotinic acid, and urea and with lower levels of sucrose; UPF intake exhibited opposite associations with these four metabolites. A higher KIDMED score was also associated with higher acetate and pantothenic acid concentrations,

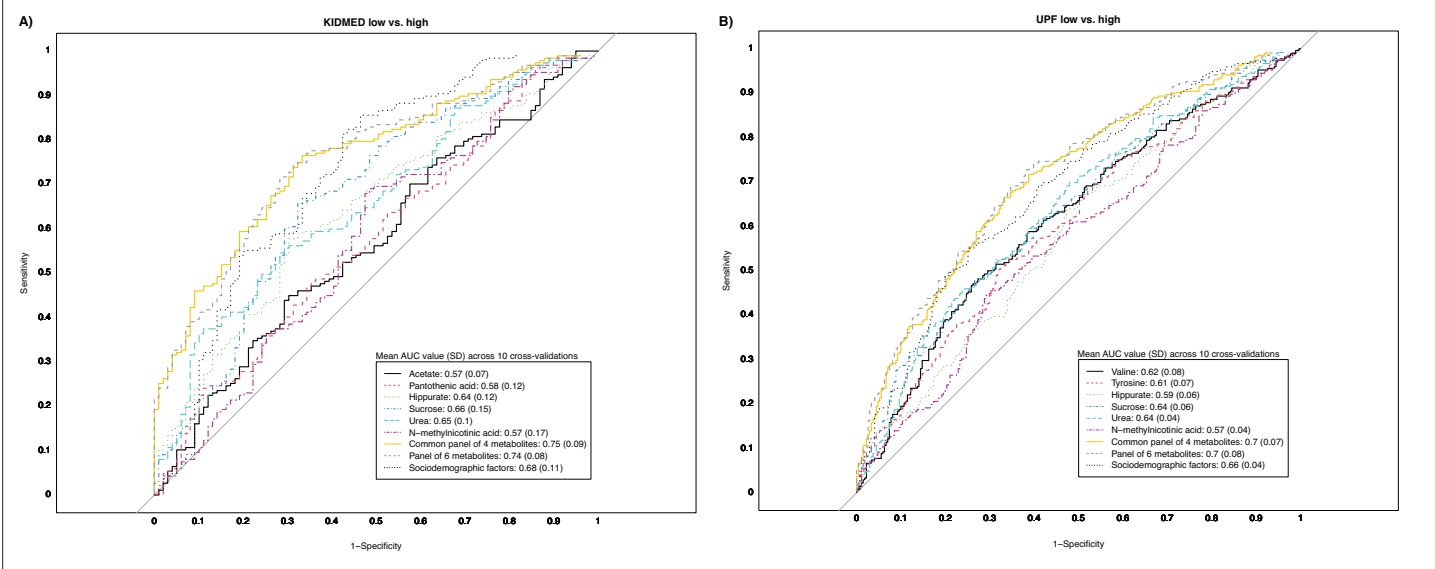

**Figure 4.** Receiver operating characteristic (ROC) curves reflecting the ability of urinary metabolites of interest in discriminating adherence to diet quality in childhood. Panel (**A**) illustrates the ability of urinary metabolites of interest in discriminating high adherence to the Mediterranean diet (KIDMED >4) from low adherence (KIDMED <1). Panel (**B**) illustrates the ability of urinary metabolites of interest in discriminating high ultra-processed food consumption (UPF ≥29% of total intake) from low consumption (UPF <18% of total daily food intake). ROC curves are based on models across the full study sample, and discriminative power is evaluated based on tenfold cross-validation. The mean area under the receiver operating characteristic curve (AUC) value (SD) across the ten cross-validations of each model is presented in the box. The common panel of four metabolites includes the metabolites associated with both diet quality indicators (hippurate, sucrose, urea, and *N*-methylnicotinid acid). The panel of six metabolites includes the metabolites associated with each diet quality indicator (common panel of four plus acetate and pantothenic acid for KIDMED, and plus valine and tyrosine for UPF). The panel of sociodemographic factors includes maternal education level, maternal prepregnancy BMI, family affluence score, child sedentary behavior, ethnicity, age, and sex.

while tyrosine and valine concentrations were inversely associated with UPF intake. There was no evidence of significant between-cohort heterogeneity in the associations between the diet quality indicators and these metabolites ( <30%; p for heterogeneity >0.2). ROC curve analyses showed that the combination of four metabolites associated with both diet quality indicators performed better than individual metabolites in discriminating children with high and low KIDMED scores and UPF intake (*Figure 4*). The discriminative ability of this metabolite panel was not improved with the addition of urinary metabolites specifically linked to each diet quality indicator and was equal or even greater to that of established sociodemographic factors previously linked to diet quality in childhood. The regression formulas (scores) for predicting children's diet quality indicators based on the urinary metabolites are given in *Supplementary file 1g*.

## Diet quality and C-peptide levels

*Table 2* presents the associations of diet quality indicators with C-peptide concentration in childhood. We found that a higher KIDMED score was associated with lower C-peptide. Specifically, compared to a low KIDMED score (<1), children with a moderate score (1–4) had a 28% lower C-peptide concentration (percent change: −27.7, 95% CI: −49.6 to 3.9) and those with a high score (>4) had a 39% lower C-peptide concentration (percent change: −39.0, 95% CI: −60.6 to −5.7) (p-trend = 0.03). An opposite association was observed for UPF intake. Compared to children at the lowest quartile of UPF intake (<18% of total daily food intake), children at the second quartile (18% to <23% of total daily food intake) had a 24% higher C-peptide concentration (percent change: 24.3, 95% CI: −6.4 to 65.2), those at the third quartile (23% to < 29% of total daily food intake) had a 39% higher concentration (percent change: 38.5, 95% CI: 3.8–84.9), and those at the fourth quartile (≥29% of total daily food intake) had a 46% higher concentration (percent change: 46.0, 95% CI: 8.1–97.3) (p-trend = 0.01). There was no evidence of interaction between the diet quality indicators (*Supplementary file 1h*). When we examined the associations separately in each cohort, we found no significant between-cohort heterogeneity ($I^2$ <18%, p for heterogeneity >0.29) (*Figure 5*). We also found no evidence that

**Table 2.** Adjusted associations of diet quality with C-peptide levels in childhood*.

| | C-peptide |
|---|---|
| | Percent change (95% CI) |
| KIDMED score (per unit increase) | −8.1 (−13.7, −2.2) |
| Low (<1) | Ref. |
| Moderate (1–4) | −27.7 (−49.6, 3.9) |
| High (>4) | −39.0 (−60.6, −5.7) |
| p-Trend | 0.03 |
| UPF intake (per 5% increase of total intake) | 9.3 (2.8, 16.2) |
| Q1 (<18% of total intake) | Ref. |
| Q2 (18% to <23% of total intake) | 24.3 (−6.4, 65.2) |
| Q3 (23% to <29% of total intake) | 38.5 (3.8, 84.9) |
| Q4 (≥29% of total intake) | 46.0 (8.1, 97.3) |
| p-Trend | 0.01 |

*Effect estimates represent percent changes in log-2 transformed C-peptide levels and their 95% CIs derived from linear regression models adjusted for maternal age, maternal education level, maternal prepregnancy BMI, family affluence status, child sex, child age, child BMI, child sedentary behavior, child ethnicity, postprandial interval, and a cohort indicator. KIDMED = Mediterranean Diet Quality Index for children and adolescents. UPF = ultra-processed food.

the observed associations differed by the sex of the children or their weight status (**Supplementary file 1i**).

## Diet-related urinary metabolites and C-peptide levels

When we examined the associations with the metabolite scores for each diet quality indicator, we found that the scores for KIDMED were associated with lower C-peptide, while opposite associations were observed with the metabolite scores for UPF (**Supplementary file 1j**). In the pairwise associations between the individual urinary metabolites linked to KIDMED or UPF intake and C-peptide, we found that higher levels of sucrose were associated with higher C-peptide levels.

## Discussion

This is the first study of European children that identified a panel of urinary metabolites associated with diet quality, as assessed by a Mediterranean diet adherence score (KIDMED) and UPF intake. These metabolites include food constituents (or their metabolic products), vitamin-related compounds and metabolites related to amino acid, protein, and carbohydrate metabolism. For both diet quality indices, KIDMED score and UPF intake, there was a common panel of four metabolites exhibiting an opposite pattern of

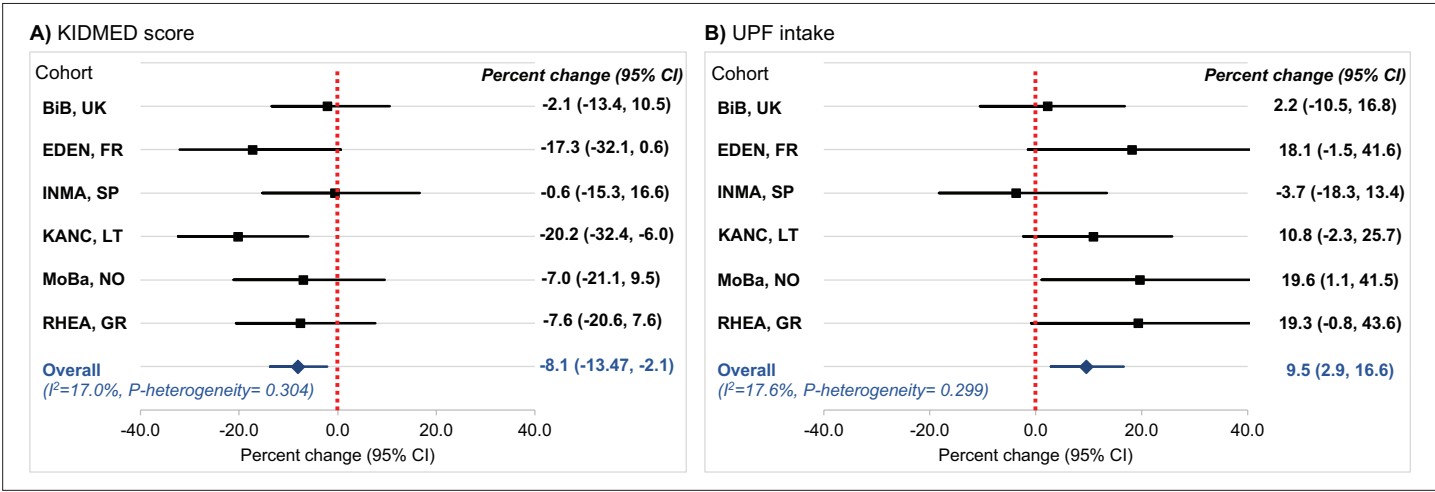

**Figure 5.** Cohort-specific associations of the diet quality indicators of interest with C-peptide in childhood. Panel (**A**) illustrates the associations for adherence to the Mediterranean diet, which was assessed via the KIDMED score (expressed per unit increase). Panel (**B**) illustrates the associations for ultra-processed food (UPF) **intake** (expressed per 5% increase of total daily food intake). Beta coefficients (95% confidence intervals, CIs) by cohort were obtained using linear regression models adjusted for maternal age, maternal education level, maternal prepregnancy body mass index (BMI), family affluence status, child sex, child age, child BMI, child sedentary behavior, child ethnicity, and postprandial interval. Combined estimates were obtained by using a fixed-effects meta-analysis. Squares represent the cohort-specific effect estimates; diamond represents the combined estimate; and horizontal lines denote 95% CIs. BiB, Born in Bradford cohort; EDEN, the Étude des Déterminants pré et postnatals du développement et de la santé de l'Enfant study; INMA, INfancia y Medio Ambiente cohort; KANC, Kaunas Cohort; MoBa, Norwegian Mother, Father and Child Cohort Study; RHEA, Rhea Mother Child Cohort study.

association: higher levels of hippurate, *N*-methylnicotinic acid, and urea, and lower levels of sucrose were concomitant with a higher KIDMED score and lower UPF intake. Moreover, these urinary metabolites that reflected these diet quality indicators, were also associated with C-peptide levels, a well-known marker of β-cell function and insulin resistance (*Patel et al., 2012*).

Previous studies assessing the metabolic response of the body to diet have largely focused on specific foods groups, *Collins et al., 2019* yet humans eat a combination of foods that may have additive or interactive effects on human physiology. In our study, we assessed two diet quality indices to better describe the complexity of overall diet. The Mediterranean diet is characterized by high intake of fruits, vegetables, legumes, nuts, and whole grain products, fish and low intakes of red meat and sweets, and it has long been appraised for its cardiometabolic benefits, even in children. (*Chatzi et al., 2017b*) Moreover, we assessed UPF intake to characterize the intake of industrial processed and prepared food items, with altered food structure, nutritional content, and taste (*Elizabeth et al., 2020*). UPF intake assessment diverts from the traditional strategy of studying nutrients, foods, or dietary patterns to identify the link between diet with health and disease, and it reflects the cumulative intake of artificial substances (i.e. flavorings, colorings, emulsifiers, and other additives) and processing byproducts. Effects of UPF consumption are likely to be the result of synergistic effects of many food ingredient compound and characteristics, and high consumption of UPF has been proposed to lead to metabolic dysregulation (*Srour et al., 2020*).

For both KIDMED and UPF, there was a common panel of four urinary metabolites (hippurate, *N*-methylnicotinic acid, urea, and sucrose), exhibiting an opposite pattern of association. In addition, KIDMED was positively associated with pantothenic acid and acetate, whereas UPF was negatively associated with two amino acids, valine and tyrosine. These metabolic signatures demonstrate a core common metabolic biomarker panel reflecting habitual dietary intake, but also show that the two diet quality indices and their metabolic signatures act complementary and can highlight different aspects of human metabolism and physiology. Our finding are consistent with previous metabolomics studies conducted in adults (*Garcia-Perez et al., 2017*; *Garcia-Perez et al., 2020*; *Almanza-Aguilera et al., 2017*). Four dietary interventions were developed (*Garcia-Perez et al., 2017*) with similar energy content and within the World Health Organization (WHO healthy eating guidelines), but with varying macro- and micronutrient intake. The urine metabolic profile for the diet most concordant with the guidelines, characterized by high intakes of dietary fiber (through fruits, vegetables, and whole grain cereal products) and low intakes of fat, sugar, and salt showed systematic differences for a total of 28 metabolites, including increased levels of acetate, hippurate, *N*-methylnicotinic acid, and urea, compared to the diet that diverted from the healthy guidelines (*Garcia-Perez et al., 2017*). The PREDIMED study also used NMR to define urinary biomarkers associated with a high adherence to a Mediterranean diet pattern in adults and the proposed biomarkers also included higher levels of urea (*Almanza-Aguilera et al., 2017*). To our knowledge, our study is the first to show the urinary metabolic footprint of total UPF intake, and also the first to show that urine NMR-derived scores of diet quality are reflective of a key biomarker of metabolic health, C-peptide, in healthy children. The ability to use a rapid, noninvasive biofluid screen to measure objective biomarkers of diet quality in children opens new avenues for exploring the significance of nutritional patterns to healthy development early in life.

Among the diet-related urinary metabolites, sucrose individually reflected the associations observed between diet quality and C-peptide (*World Health Organization, 2015*). Elevated levels of added sugars could lead to elevated glucose load in the human body, which in turn can lead to an increase in glycemic response and, thus, C-peptide production in healthy populations. To the best of our knowledge, there is only one previous study from Mexico examining the relation of diet to C-peptide in children (*Perng et al., 2017*). Similar to our findings, this study showed that adherence to a prudent dietary pattern characterized by food groups commonly consumed in Mediterranean diet (vegetables, fruit, fish, and legumes) was associated with low C-peptide levels in boys. Regarding other markers of glucose regulation, our findings are in line with previous studies in children and adults reporting that low adherence to a healthy dietary pattern or high consumption of specific UPFs (sugar-sweetened beverages and ultra-processed meat) were associated with impaired glucose regulation and insulin resistance (*Manios et al., 2010*; *Asghari et al., 2016*; *Chan She Ping-Delfos et al., 2015*; *McKeown et al., 2018*; *Ley et al., 2014*; *Fiorito et al., 2009*). Overall, our findings are consistent with the public health concerns raised by the WHO, relating poor overall quality with high intake level of sugars and

added sugars, and with poor metabolic health and a high risk of several noncommunicable diseases (*World Health Organization, 2015*; *Stanhope et al., 2013*; *WHO Study Group, 1990*; *WHO, 2003*).

Regarding the other urinary metabolites reflecting diet quality, hippurate is a normal component of urine, a metabolic product of phenolic compounds which are present in various dietary sources. It is also a biomarker of fruit/vegetable intake, as it has been confirmed in previous studies in healthy children, adolescents (*Lau et al., 2018*; *Krupp et al., 2012*) and adults (*Edmands et al., 2011*) *N*-methylnicotinic acid (trigonelline) is a product of the metabolism of niacin (vitamin B3) and a biomarker of various dietary sources like legumes (*Sri Harsha et al., 2018*) and fruits (*Lau et al., 2018*). Urea is the principal end product of amino acids and protein catabolism and a potential marker of protein intake from food (*Garcia-Perez et al., 2020*).

In addition, adherence to the Mediterranean diet was also positively associated with urinary levels of pantothenic acid and acetate. Both compounds have a central role in human biochemistry and the metabolism and synthesis of carbohydrates, proteins, and fats. Pantothenic acid (vitamin B5, necessary to form coenzyme-A) is present in many foods, and we have previously reported a positive association between consumption of dairy products and urinary pantothenic acid in the same study population (*Lau et al., 2018*). Further, we have previously shown that BMI is negatively associated with urinary levels of this metabolite (*Lau et al., 2018*) and our results suggest that higher adherence to the Mediterranean diet associates with pantothenic acid independently of the potential influence of BMI. Acetate has multiple dietary sources, it is produced by acetate-producing bacteria in foodstuff and urinary acetate is also modulated by human gut microbial metabolism. In our previous HELIX analysis on specific food group intakes, we have shown a positive association between potato consumption and urinary acetate levels (*Lau et al., 2018*).

Moreover, we found that UPF intake was negatively associated with two urinary amino acids, valine and tyrosine. Tyrosine is regarded as a conditionally essential amino acid in adults and essential in children. Foods high in dietary tyrosine include dairy, meat, eggs, beans, nuts, grains. Tyrosine is a precursor for neurotransmitters and hormones, increases dopamine availability which in turn could enhance cognitive performance (*Kühn et al., 2019*). Valine is an essential branch chain amino acid (BCAA) critical to energy homeostasis, protein and muscle metabolism (*Brosnan and Brosnan, 2006*; *Nie et al., 2018*). In many studies, it has been observed that elevated BCAAs are associated with insulin resistance and diabetes (*Lynch and Adams, 2014*). Also, in our previous HELIX study, *Lau et al., 2018* we found that urinary valine was associated with higher children's BMI. However, it remains to be elucidated whether these associations are causal (e.g., via mTOR activation) or consequential (e.g., due to reduced mitochondrial oxidation) in metabolic disease, (*Lynch and Adams, 2014*) and whether UPF intake plays a role in the etiology of the association of BCAAs with metabolic health.

## Strengths and limitations

The main strengths of the study are the multicentric design which included children from six countries spanning north to south in Europe, the use of standardized data collection and biomarker measurement protocols across cohorts, and the fairly large sample size with biomarker data. Identified panels of urinary metabolites had similar or higher capacity in discriminating children's diet quality to that of established sociodemographic determinants. We chose [1]H NMR spectroscopy for urinary analysis as this is an inherently high reproducible, high throughput technique suitable for the identification and quantification of urine metabolites, which are typically of high concentrations, without complex sample preparation which could potentially introduce analytical biases. Also, urinary [1]H NMR spectroscopy has been applied in many other cohort studies and proposed as an objective method for dietary assessment (*Garcia-Perez et al., 2017*) potentially facilitating comparative studies in the future. We have used a pooled urine sample collection design which combined the last sample before bedtime with the first morning void sample of the following day, and we have shown in our preliminary work that this sample collection strategy has the advantage of reducing diurnal variations (*Lau et al., 2018*; *Maitre et al., 2017*).

Our study has also some limitations. As in any observational study, there is the possibility of unmeasured residual confounding. In our analyses, we took into account a number of sociodemographic and lifestyle factors in childhood (e.g., socioeconomic status, ethnicity) that are associated with both diet quality and glycemic response. We did not have data available on children's physical activity. Nevertheless, we adjusted all our models for sedentary behavior (including time spent in front of the

screen) which has been shown to associate to physical activity levels, as the time devoted to sedentary screen-time activities might affect availability of time devoted for exercise, or vice versa (*Serrano-Sanchez et al., 2011*; *Pearson et al., 2014*; *Aira et al., 2021*) Further, we did not have data available to control for energy intake. However, in all our models, we included BMI of the children, a measure strongly correlated to energy intake, (*Jakes et al., 2004*) and assessed UPF intake as proportion of total food intake. Moreover, the absence of heterogeneity across cohorts with different correlation structures and confounding patterns in their data (*Maitre et al., 2018*), provide evidence to support that unmeasured confounding is unlikely to have influenced the observed associations. Since the data collected are cross-sectional, and there is no temporality in the observed associations, further longitudinal studies examining metabolic and glycemic alterations in relation to diet quality are needed. Although $^1$H NMR spectroscopy had the advantage of improving the specificity of the quantitation and provided explicit metabolite identification, it limited the number of metabolites being measured and provided partial coverage of the urine metabolome. Absolute quantification of some metabolites with exchangable protons such as urea could also be negatively impacted by the solvent suppression methods required for $^1$H NMR spectroscopy of urine. Supplementing the current study with other complementary untargeted and targeted metabolomic approaches in future, such as mass spectrometry, would help enhance identification and robust quantification of urinary metabolites associated with diet quality in children.

In summary, this multicenter European study showed that urinary metabolic profiles related to food constituents (or their metabolic products), to amino acid and carbohydrate metabolism reflect adherence to the Mediterranean diet and UPF intake in childhood. Higher adherence to Mediterranean diet, lower UPF intake, and lower levels of the diet-related carbohydrate sucrose were associated with lower C-peptide levels, a marker of β-cell function. These results provide evidence to support efforts by public health authorities to recommend increased adherence to the Mediterranean diet and limiting UPF consumption in childhood. Further prospective studies examining the association of diet quality and related metabolomic profiles with C-peptide and other surrogates of insulin resistance are needed to replicate our findings.

## Acknowledgements

We acknowledge the input of the entire HELIX consortium. We are grateful to all the participating families in the five cohorts (BiB, EDEN, INMA, MoBa, KANC, and RHEA cohorts), that took part in this study. We are equally grateful to all the fieldworkers for their dedication and efficiency in this study. A full roster of the INMA and RHEA study investigators can be found here and here, respectively. The Born in Bradford study is only possible because of the enthusiasm and commitment of the participating children and parents. We are grateful to all the participants, health professionals, and researchers who have made Born in Bradford happen. We are also grateful to all the participating families in Norway who take part in the ongoing MoBa cohort study. We thank all the children and families participating in the EDEN-HELIX mother–child cohort. We are grateful to Joane Quentin, Lise Giorgis-Allemand, and Rémy Slama (EDEN study group) for their work on the HELIX project. We thank Sonia Brishoual, Angelique Serre, and Michele Grosdenier (Poitiers Biobank, CRB BB-0033-00068, Poitiers, France) for biological sample management and Prof Frederic Millot (principal investigator), Elodie Migault, Manuela Boue, and Sandy Bertin (Clinical Investigation Center, Inserm CIC1402, CHU de Poitiers, Poitiers, France) for planification and investigational actions. We are also grateful to Veronique Ferrand-Rigalleau, Celine Leger, and Noella Gorry (CHU de Poitiers, Poitiers, France) for administrative assistance. We also acknowledge the commitment of the members of the EDEN Mother-Child Cohort Study Group: I Annesi-Maesano, JY Bernard, J Botton, MA Charles, P Dargent-Molina, B de Lauzon-Guillain, P Ducimetière, M de Agostini, B Foliguet, A Forhan, X Fritel, A Germa, V Goua, R Hankard, M Kaminski, B Larroque, N Lelong, J Lepeule, G Magnin, L Marchand, C Nabet, F Pierre, MJ Saurel-Cubizolles, M Schweitzer, and O Thiebaugeorges. No external funding was received for this work. The HELIX project has received funding from the European Community's Seventh Framework Programme (FP7/2007–2013) under grant agreement no. 308,333. The STOP project (http://www.stopchildobesity.eu/) received funding from the European Union's Horizon 2020 research and innovation programme under grant agreement no. 774,548. The STOP Consortium is coordinated by Imperial College London and includes 24 organizations across Europe, the United States, and New Zealand. The content of this publication reflects only the views of the authors, and

the European Commission is not liable for any use that may be made of the information it contains. INMA data collections were supported by grants from the Instituto de Salud Carlos III, CIBERESP, and the Generalitat de Catalunya-CIRIT. KANC was funded by the grant of the Lithuanian Agency for Science Innovation and Technology (6-04-2014_31 V-66). For a full list of funding that supported the EDEN cohort, refer to: Heude B et al. Cohort Profile: The EDEN mother–child cohort on the prenatal and early postnatal determinants of child health and development. Int J Epidemiol. 2016 Apr;45(2):353–63. The Norwegian Mother, Father and Child Cohort Study (MoBa) is supported by the Norwegian Ministry of Health and Care Services and the Ministry of Education and Research. The Rhea project was financially supported by European projects, and the Greek Ministry of Health (Program of Prevention of Obesity and Neurodevelopmental Disorders in Preschool Children, in Heraklion district, Crete, Greece: 2011–2014; 'Rhea Plus': Primary Prevention Program of Environmental Risk Factors for Reproductive Health, and Child Health: 2012–2015). Born in Bradford received funding from the Wellcome Trust (101597). Professor Wright and McEachan receive funding from the National Institute for Health Research Applied Research Collaboration for Yorkshire and Humber. The views expressed are those of the author(s) and not necessarily those of the NIHR or the Department of Health and Social Care. Dr. Maribel Casas received funding from Instituto de Salud Carlos III (Ministry of Economy and Competitiveness) (MS16/00128). Dr. Leda Chatzi was supported by NIH/NIEHS R01 ES029944, R01ES030691, R01ES030364, R21 ES029681, R21 ES028903, and P30 ES007048-23. Dr. David Conti was supported by P01CA196569, R01CA140561, R01 ES016813, R01 ES029944, R01ES030691, and R01ES030364. Dr. Nikos Stratakis was supported by NIH/NIEHS R21 ES029681 and P30 ES007048-23, and NIH/NIDDK P30 DK048522-24. Dr. Hector Keun and Dr. Alexandros Siskos were also supported by the European Union's Horizon 2020 research and innovation programme under grant agreement no. 874,583 ('ATHLETE'). Dr. Eleni Papadopoulou was supported by the Research Council of Norway, under the MILJØFORSK program (project no. 268465). Dr. Fernanda Rauber was supported by the Fundação de Amparo à Pesquisa do Estado de São Paulo 2016/14302-7 and 2018/19820-1. The CRG/UPF Proteomics Unit is part of the Spanish Infrastructure for Omics Technologies (ICTS OmicsTech) and it is a member of the ProteoRed PRB3 consortium which is supported by grant PT17/0019 of the PE I + D + i 2013–2016 from the Instituto de Salud Carlos III (ISCIII) and ERDF. ISGlobal acknowledges support from the Spanish Ministry of Science, Innovation and Universities, 'Centro de Excelencia Severo Ochoa 2013–2017', SEV-2012–0208, and 'Secretaria d'Universitats i Recerca del Departament d'Economia i Coneixement de la Generalitat de Catalunya' (2017SGR595).

## Additional information

### Competing interests

Muireann Coen: Is an employee of AstraZeneca. The author declares that no other competing interests exist. The other authors declare that no competing interests exist.

### Funding

| Funder | Grant reference number | Author |
|---|---|---|
| European Community's Seventh Framework Programme (FP7) | 308333 | Regina Grazuleviciene John Wright Martine Vrijheid Hector C Keun |
| European Union's Horizon 2020 | 774548 | Christopher Millett Paolo Vineis |
| National Institutes of Health (NIH)/National Institute of Environmental Health Sciences (NIEHS) | R21ES02968 | Nikos Stratakis David V Conti Leda Chatzi |
| National Institute for Health Research Applied Research Collaboration for Yorkshire and Humber | | Rosemary RC McEachan John Wright |

| Funder | Grant reference number | Author |
|---|---|---|
| Instituto de Salud Carlos III | MS16/00128 | Maribel Casas |
| Research Council of Norway, under the MILJØFORSK program | 268465 | Eleni Papadopoulou |
| Fundação de Amparo à Pesquisa do Estado de São Paulo | 2016/14302-7 and 2018/19820-1 | Fernanda Rauber |
| Instituto de Salud Carlos III (ISCIII) and ERDF | PT17/0019 of the PE I+D+i 2013-2016 | Eduard Sabidó |
| NIH/NIEHS | R01ES029944 | Leda Chatzi David V Conti |
| NIH/NIEHS | R01ES030691 | Leda Chatzi David V Conti |
| NIH/NIEHS | R01ES030364 | Leda Chatzi David V Conti |
| NIH/NIEHS | P30ES007048 | Leda Chatzi David V Conti Nikos Stratakis |
| NIH | P01CA196569 | David V Conti |
| NIH | R01CA140561 | David V Conti |
| NIH | R01ES016813 | David V Conti |
| NIH | P30DK048522 | Nikos Stratakis |
| Departament de Salut de la Generalitat de Catalunya | Spanish regional program PERIS (Ref.: SLT017/20/000119) | Jose Urquiza |

The funders had no role in study design, data collection, and interpretation, or the decision to submit the work for publication.

## Author contributions

Nikos Stratakis, Conceptualization, Data curation, Formal analysis, Investigation, Methodology, Visualization, Writing – original draft, Writing – review and editing; Alexandros P Siskos, Conceptualization, Data curation, Investigation, Methodology, Writing – original draft, Writing – review and editing; Eleni Papadopoulou, Conceptualization, Formal analysis, Investigation, Methodology, Writing – original draft, Writing – review and editing; Anh N Nguyen, Formal analysis, Investigation, Methodology, Writing – review and editing; Yinqi Zhao, Katerina Margetaki, Formal analysis, Investigation, Writing – review and editing; Chung-Ho E Lau, Data curation, Methodology, Writing – review and editing; Muireann Coen, Data curation, Investigation, Writing – review and editing; Lea Maitre, Data curation, Investigation, Project administration, Writing – review and editing; Silvia Fernández-Barrés, Lydiane Agier, Anne Lise Brantsaeter, Investigation, Methodology, Writing – review and editing; Sandra Andrusaityte, Serena Fossati, Barbara Heude, Helle Margrete Meltzer, Christopher Millett, Fernanda Rauber, Theano Roumeliotaki, Eva Borras, Eduard Sabidó, Marina Vafeiadi, Paolo Vineis, David V Conti, Investigation, Writing – review and editing; Xavier Basagaña, Funding acquisition, Investigation, Methodology, Writing – review and editing; Maribel Casas, Regina Grazuleviciene, Rosemary RC McEachan, John Wright, Martine Vrijheid, Funding acquisition, Investigation, Writing – review and editing; Oliver Robinson, Jose Urquiza, Investigation, Project administration, Writing – review and editing; Trudy Voortman, Methodology, Writing – review and editing; Hector C Keun, Leda Chatzi, Conceptualization, Funding acquisition, Investigation, Methodology, Supervision, Writing – original draft, Writing – review and editing

## Author ORCIDs

Nikos Stratakis http://orcid.org/0000-0003-4613-0989
Alexandros P Siskos http://orcid.org/0000-0002-5635-7426
Anne Lise Brantsaeter http://orcid.org/0000-0001-6315-7134

## Ethics

Prior to the start of HELIX, all six cohorts on which HELIX is based had undergone the required evaluation by national ethics committees and had obtained all the required permissions for their cohort recruitment and follow-up visits. Each cohort also confirmed that relevant informed consent and approval were in place for secondary use of data from preexisting data. The work in HELIX was covered by new ethics approvals from the local ethics committees at each site, and at enrolment in the HELIX subcohort, participants were asked to sign an informed consent form for the specific HELIX work including clinical examination and biospecimen collection and analysis. Additionally, the current study (study ID: HS-20-00390) was approved by the University of Southern California Institutional Review Board.

## Decision letter and Author response

Decision letter https://doi.org/10.7554/eLife.71332.sa1
Author response https://doi.org/10.7554/eLife.71332.sa2

---

# Additional files

## Supplementary files

• Supplementary file 1. Additional results. (a) KIDMED items, and their scoring, used to assess adherence to the Mediterranean diet in HELIX children. (b) Items included in the FFQ by ultra-processed food (UPF) inclusion and information regarding the extent and purpose of food processing. (c) Intakes of food groups (in servings/week) by categories of the KIDMED score. (d) Intakes of food groups (in servings/week) by quartiles of ultra-processed food intake. (e) Associations of KIDMED score with urinary metabolites in childhood. (f) Associations of ultra-processed food consumption with urinary metabolites in childhood. (g) Regression formulas (scores) for predicting diet quality in childhood based on panels of urinary metabolites. (h) Interaction of diet quality indicators in association to C-peptide concentration in childhood. (i) Associations of diet quality with C-peptide concentration in childhood after stratifying by sex and by weight status, respectively. (j) Associations between urinary metabolites linked to diet quality and C-peptide in childhood.

• Transparent reporting form

## Data availability

Due to the HELIX data policy and data use agreement, human subjects data used in this project cannot be freely shared. Researchers external to the HELIX Consortium who have an interest in using data from this project for reproducibility or in using data held in general in the HELIX data warehouse for research purposes can apply for access to data for a specific manuscript at the time. Interested researchers should fill in the application protocol found in ANNEX I at https://www.projecthelix.eu/files/helix_external_data_request_procedures_final.pdf and send this protocol to helixdata@isglobal.org. The applications are received by the HELIX Coordinator, and are processed and approved by the HELIX Project Executive Committee. The decision to accept or reject a proposal is taken by the HELIX Project Executive Committee. This decision will be based largely on potential overlap with other HELIX-related work, the adequacy of data protection plans, and the adequacy of authorship and acknowledgement plans. After approval by the HELIX PEC, the cohorts participating in HELIX will each be asked if they approve use of their data in the proposal and what their conditions are for participation. Each cohort will have the opportunity to opt out at this stage if any of their cohort-specific conditions are not met. These conditions are to be stated clearly by the cohort at this stage and the cohort should be open to discussing how these conditions could be fulfilled in subsequent proposal revisions. Further details on the content of the data warehouse (data catalog) including those data used for the present projects and procedures for external access are described on the project website http://www.projecthelix.eu/index.php/es/data-inventoryhttp://www.projecthelix.eu/index.php/es/data-inventory. Code used for data analysis has been described with references in the methods section under Statistical analysis. Source data for Figure 1 are provided in Supplementary File 1.

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
