## [Editor Report]

This well executed study looks at the association of urinary metabolites to the types of diets consumed by European children. Using NMR they find four metabolites that are predictive of a Mediterranean diet. This presents both an approach additional to traditional questionnaire methods and potential insights into biological pathways and will be of interest to nutritionists and epidemiologists.

---

## [Decision Letter]

**Decision letter after peer review:**

Thank you for submitting your article "Urinary metabolic biomarkers of diet quality in European children are associated with metabolic health" for consideration by *eLife*. Your article has been reviewed by 1 peer reviewer and the evaluation has been overseen by a Reviewing Editor and Martin Pollak as the Senior Editor. The reviewer has opted to remain anonymous.

The reviewer has discussed their review with the Reviewing Editor who has drafted this to help you prepare a revised submission.

Essential revisions:

1) The one omission is the effects of activity levels and total caloric consumption. There is an attempt to link body weight to C-peptide associations, but in a revision, it would be nice to also include MBI as a parameter for the concentrations of metabolites.

*Reviewer #1 (Recommendations for the authors):*

The only additional recommendation from the public review is to rework figure one to make it easier to follow the metabolites per spiral plot. Following them down to the inner circle is more difficult than it needs to be.

For supplemental figures 3 and 4, the abbreviations for the cohorts needs to be referenced.

---

## [Author Response]

Reviewer #1 (Recommendations for the authors):The only additional recommendation from the public review is to rework figure one to make it easier to follow the metabolites per spiral plot. Following them down to the inner circle is more difficult than it needs to be.

We have modified the figure depicting the diet-metabolite associations (now Figure 3) to facilitate readership. Please note that the numbering of all figures have been changed as all figures previously included as supplementary material have now been moved to the main text to adhere to the journal’s guidelines.

For supplemental figures 3 and 4, the abbreviations for the cohorts needs to be referenced.

We have now referenced the abbreviations for the cohorts in all relevant figures. These are now figures 5A and 5B. Please note that the numbering of all figures have been changed as all figures previously included as supplementary material have now been moved to the main text to adhere to the journal’s guidelines.

Modified text in figure:

Figure 5. Cohort-specific associations of the diet quality indicators of interest with C-peptide in childhood. Panel A illustrates the associations for adherence to the Mediterranean diet, which was assessed via the KIDMED score (expressed per unit increase). Panel B illustrates the associations for ultra-processed food (UPF) intake (expressed per 5% increase of total daily food intake). Β coefficients (95% CIs) by cohort were obtained using linear regression models adjusted for maternal age, maternal education level, maternal pre-pregnancy BMI, family affluence status, child sex, child age, child BMI, child sedentary behavior, child ethnicity, and postprandial interval. Combined estimates were obtained by using a fixed-effects meta-analysis. Squares represent the cohort-specific effect estimates; diamond represents the combined estimate; and horizontal lines denote 95% CIs. BiB, Born in Bradford cohort; EDEN, the Étude des Déterminants pré et postnatals du développement et de la santé de l’Enfant study; INMA, INfancia y Medio Ambiente cohort; KANC, Kaunas Cohort; MoBa, Norwegian Mother, Father and Child Cohort Study; RHEA, Rhea Mother Child Cohort study.

References:

1. Serrano-Sanchez JA, Marti-Trujillo S, Lera-Navarro A, Dorado-Garcia C, Gonzalez-Henriquez JJ, Sanchis-Moysi J. Associations between screen time and physical activity among Spanish adolescents. PLoS One. 2011;6(9):e24453.

2. Pearson N, Braithwaite RE, Biddle SJ, van Sluijs EM, Atkin AJ. Associations between sedentary behaviour and physical activity in children and adolescents: a meta-analysis. Obes Rev. 2014;15(8):666-675.

3. Aira T, Vasankari T, Heinonen OJ, et al. Physical activity from adolescence to young adulthood: patterns of change, and their associations with activity domains and sedentary time. Int J Behav Nutr Phys Act. 2021;18(1):85.

4. Lau CE, Siskos AP, Maitre L, et al. Determinants of the urinary and serum metabolome in children from six European populations. BMC Med. 2018;16(1):202.

5. Jakes RW, Day NE, Luben R, et al. Adjusting for energy intake--what measure to use in nutritional epidemiological studies? Int J Epidemiol. 2004;33(6):1382-1386.

6. Kühn S, Düzel S, Colzato L, et al. Food for thought: association between dietary tyrosine and cognitive performance in younger and older adults. Psychological Research. 2019;83(6):1097-1106.

7. Brosnan JT, Brosnan ME. Branched-Chain Amino Acids: Enzyme and Substrate Regulation. The Journal of Nutrition. 2006;136(1):207S-211S.

8. Nie C, He T, Zhang W, Zhang G, Ma X. Branched Chain Amino Acids: Beyond Nutrition Metabolism. Int J Mol Sci. 2018;19(4).

9. Lynch CJ, Adams SH. Branched-chain amino acids in metabolic signalling and insulin resistance. Nat Rev Endocrinol. 2014;10(12):723-736.